# The Effect of Cinnamon and Ginger Spices on Anthocyanins in Sweetened Roselle Beverages

Esereosa D. Omoarukhe [1,*,†], Niamh Harbourne [2] and Paula Jauregi [1,‡,§]

1    Department of Food and Nutritional Sciences, University of Reading, Harry Nursten Building, Reading RG6 6DZ, UK
2    UCD Institute of Food and Health, School of Agricultural and Food Science, University College Dublin, Belfield, D04 V1W8 Dublin, Ireland
*    Correspondence: dilenu@yahoo.com
†    Current address: Pharmatech Labs, 1352 West 300 South, Lindon, UT 84042, USA.
‡    Current address: AZTI, Food Research, Basque Research and Technology Alliance (BRTA), Parque Tecnlógico de Bizkaia, Astondo Bidea, Edificio 609, Derio, 48160 Bizkaia, Spain.
§    Current address: Ikerbasque, Basque Foundation for Science, 48013 Bilbao, Spain.

**Abstract:** This study explores the potential benefits of spices (cinnamon and ginger) on Roselle anthocyanins within a sweetened Roselle beverage matrix. Anthocyanins and other related properties of the beverage (colour, antioxidant capacity, total phenolics, and pH) were observed from the start and monitored for 30 days at accelerated storage conditions (40 °C). The sweeteners at the amounts used (80 g/L granulated sugar and 0.32 g/L Stevia Reb A) did not have a significant effect on the initial anthocyanin content in the beverage and did not significantly impact degradation. Upon the addition of spices to the sweetened beverage, ginger (1 g/L) did not result in significant changes, initially or during storage. However, following the addition of cinnamon (1 g/L) to the beverages (unsweetened and sweetened), an initial increase in the total phenolic and FRAP antioxidant activity in the Roselle beverages was observed; furthermore, it reduced the degradation of anthocyanins and improved colour stability during storage. This effect is postulated to be due to a co-pigmentation reaction or the acylation of anthocyanins with a complex formed from the reaction of glucose with the phenolic compounds contained in cinnamon.

**Keywords:** Roselle beverage; anthocyanin; stability; spices; Stevia rebaudioside A

## 1. Introduction

Roselle extracts, the base ingredient in Roselle beverages, are rich in polyphenolic compounds. Polyphenolic compounds are bioactive compounds consisting of aromatic rings attached to one or more hydroxyl groups [1]. Roselle polyphenols consist of flavonoids including anthocyanins, protocatechuic acid, quercetin [2], gossypetin, hibiscetrin [3], hibiscetine, and sabdaretine, phytosterols [3], eugenol alongside some suspected compounds, such as gossypin, gossytrin, rutin, isoquercitrin, kaempferol 3-rhamnoglucoside, kaempferol 3-gluco side, cannabiscitrin, and myricetin [4]. However, anthocyanins are the most interesting of all the polyphenols in Roselle extracts; mainly including delphinidin 3-O-sambubioside and cyanidin 3-sambubioside [5,6]. Although some other studies have also reported delphinidin 3-glucosides and cyanidin 3-glucosides as minor anthocyanins [7]. In general, anthocyanins are bioactive compounds that provide both the medicinal and sensory properties associated with a plant [1]. The content of anthocyanins in Roselle was reported to be in the range of 1.5–2.5 g/100 g dry weight [6,7]. However, it is well known that anthocyanins are unstable and typically degrade with increased water activity, changes in pH, temperature, oxygen, and light [8].

Although sugar is primarily added to beverages to improve palatability [9], it may also function as a stabilising agent for Roselle anthocyanins. The aglycone form, anthocyanidin, which consists of a hydroxyl flavylium base structure, becomes more stable when

linked to sugars, usually glucose [1]. However, a high concentration of sugar is expected to control water activity and, consequently, prevent the hydration of the flavylium ring, which would lead to the deterioration of the anthocyanin. Kopjar and Piližota [10] demonstrated that sugars such as glucose and trehalose (at 10% of formula) provide stabilising effects on anthocyanins.

The potential health benefits of the Roselle beverage, for instance, its antidiabetic properties [11,12], could be hindered with the inclusion of sucrose (henceforth referred to as sugar) in the beverage. Although there are several low-calorie intensity sweeteners available to replace sugar, only Stevia glycosides currently satisfy consumer clean label demands and have been approved for use across continents including the EU and North America. One particularly pertinent study was carried out by Woźniak, Marszałek, and Skąpska [13] on model buffer solutions, in which it was observed that 0.05–0.2 mg/L steviol glycosides (stevioside and rebaudioside A) had no effect on the degradation of anthocyanins upon storage, whereas 50–200 g L$^{-1}$ sugars (glucose, fructose, and sucrose) increased the stability of anthocyanins (cyanidin-3-glucoside and pelargonidin-3-glucoside). Notwithstanding, further investigations on stabilising the potential of both sugar and Stevia Reb A on anthocyanins in more typical beverage systems are necessary.

In addition to sweeteners, a more complex beverage system would contain natural flavours, such as juices or fruits, vegetables, and spice extracts, which due to their phenolic compositions, may further affect anthocyanin stability. Of these natural flavourings, spices are the least studied in beverages, although spices are in use in contemporary ready-to-drink sweetened beverages, such as ginger ale, ginger beer, root beer, lassi, spiced elderberry infusions, mulled wine, chicha morada (Peruvian) alongside a range of herbal or spicy infusions, which are available internationally. Moreover, two very recent studies [14,15] explored the use of spices (i.e., basil and ginger) to improve the antioxidant properties of Roselle beverages. Indeed, Abidoye and co-workers [14] successfully increased antioxidant activity when using basil (5–15% of the mixture).

Two spices have been selected for this study due to their popularity in the beverage industry and their congruency with Roselle beverages, namely, Cinnamomum zeylanicum (cinnamon) and Zingiber officinale (ginger). Both spices have good antioxidant activity [16,17] although cinnamon has about four times the activity of ginger [18]. Nevertheless, limited information exists on the effects of either of these spices on anthocyanins.

The objectives of this present study are to firstly investigate the effect of cinnamon and ginger on anthocyanin and related properties of sweetened Roselle beverages during storage and secondly, the possible interactions between the spices, sweeteners, and anthocyanins during the storage of the Roselle beverages.

## 2. Materials and Methods

### 2.1. Beverage Preparations

Five grams of commercial air-dried Roselle (Just Ingredients, Wotton-Under-Edge, UK) was steeped in 1 litre of water at 90 °C for 25 min in a Duran flask and shaken (86 rev/min) in a water bath (Grant OLS 200). The extract was filtered under a vacuum with Whatman filter paper (No. 4), then, cooled on ice. The extract was divided into three portions and one portion was left unsweetened (control), while the other portions were sweetened with either white granulated sugar (80 g/L) or Stevia rebaudioside A (0.32 g/L; 80% purity—Cargill, UK), henceforth, referred to as 'stevia'. Each of the three extracts was further split into three portions and flavoured with ground dried spice powder: 1 g/L cinnamon and 1 g/L ginger (Just Ingredients, UK), leaving one portion without spice. These were refrigerated overnight (18 h) and filtered as above. The resulting 9 products were stored at 40 °C for 30 days, with the analysis carried out on aliquots every day for the first 5 days; 2–3 days for the next 10 days; and from then on, every 5 days until day 30.

*2.2. Real-Time Storage*

Real-time storage studies were carried out prior to this experimental work. Some of the results are detailed in a related study by Omoarukhe [19]. In relation to this study, control (unsweetened) and sweetened (with sugar and stevia) Roselle beverages were stored at room temperature (21 °C) for 90 days and analysed every 10 days until day 30, and then monthly until 90 days. A summary of the kinetic data from the initial real-time study on the changes in anthocyanin profiles for sweetened and unsweetened Roselle beverages is provided in Section 3.1.

*2.3. Chemical Kinetics*

The natural logarithms of the concentration of total monomeric anthocyanins were plotted against time (in days) to confirm adherence to the first-order kinetics (Equation (5)). A linear trend was obtained for almost all samples ($R^2$ range was 0.977–0.991) and the linear equation was used to obtain the reaction rate constant ($k$).

$$Ln \, \frac{A_o}{A} = kt \tag{1}$$

where $A_o$ is the initial concentration and $A$ the concentration at a given time.

The half-life ($t_{1/2}$), which is the time taken for anthocyanins to reach half their initial value, was determined by applying Equation (6), derived from the first-order reaction equation (Equation (5)):

$$t_{\frac{1}{2}} = \frac{ln2}{k} \tag{2}$$

The equivalent days of real-time storage for the accelerated storage were obtained by comparing the rate of reactions between the real-time storage and the accelerated storage in this current study, for each parameter. Assuming the initial concentration ($A_o$) and the final concentration ($A_e$) remain unchanged for real-time and accelerated conditions, then, the first order equation could be re-written as:

$$Ln\left(\frac{A_o}{A_e}\right) = kt_s = k_a t_a \tag{3}$$

where $t_s$ is the real-time shelf life; k is the rate constant in real time; $t_a$ is the accelerated shelf life; and $k_a$ is the accelerated rate constant. Therefore, for 1 day of accelerated study, the real storage time equivalent for the relevant parameters can be calculated as:

$$t_s = \frac{k_a}{k} \tag{4}$$

*2.4. Accelerated Storage*

The conditions: 40 °C for 30 days were selected for this study. Aliquots of Roselle beverages from Section 2.1 were stored under accelerated conditions at 40 °C for 30 days in a stability cabinet (Sanyo Gallenkamp, Loughborough, UK).

*2.5. Analysis of Samples*

2.5.1. Chemical Tests

Total Monomeric Anthocyanin

Total monomeric anthocyanins were determined using the pH differential method [20]; the extract was mixed individually with pH 1.0 or 4.5 pH buffer solutions in a ratio of 1:4 and left for 20 min. The absorbance of the test portions at pH 1.0 and 4.5 was determined spectrophotometrically (Amersham Pharmacia Biotech Ultrospec 1100 pro UV spectrophotometer) at 520 nm and 700 nm, respectively. Anthocyanin pigment concentrations were

expressed in cyanidin-3-glucoside (C3G) equivalents. Calculations were carried out using the following equation:

$$⟦Anthocyanin\ pigment\ \left(C3G, \frac{mg}{L}\right) = \left(A \times MW \times DF \times 10⟧^3\right) / (\mathcal{E} \times 1) \tag{5}$$

where $A = (A_{520nm} - A_{700nm})_{pH\ 1.0} - (A_{520nm} - A_{700nm})_{pH\ 4.5}$; MW (molecular weight) = 449.2 g/mol for cyanidin-3-glucoside; DF = dilution factor; 1 = pathlength in cm; $\varepsilon$ = 26,000 molar extinction coefficients in $L \times mol^{-1} \times cm^{-1}$ for cyanidin-3-glucoside, and $10^3$ = factor for conversion from g to mg, and cm.

Total Phenolic Content

Folin–Ciocalteu (FC) colorimetry was used to determine the total phenolic contents in the extracts. Extract/standard (0.2 mL) was added to 6.0 mL of distilled water in 10 mL volumetric flasks, after which 0.5 mL Folin–Ciocalteu reagent (Sigma-Aldrich, Darmstadt, Germany) was added and mixed. After 1 min and before 8 min, 1.5 mL of 20% sodium carbonate (Fisher Scientific, UK) solution was also added, and the volume was adjusted to 10 mL with water. The colour generated after 2 h was read at 760 nm using an Amersham Pharmacia Biotech Ultrospec 1100 pro UV spectrophotometer. Gallic acid (Sigma-Aldrich, Darmstadt, Germany) standards, with concentrations ranging from 0.05 to 1 g/L, were used to generate standard plots and an equation for the calculation of the total phenolic concentration in each extract. The results were expressed in milligrams of gallic acid equivalent per litre (mg GAE/L)

Antioxidant Capacity

Antioxidant capacity was measured using the ferric reducing antioxidant power (FRAP) assay, originally proposed by [21], although with some modifications. This method was selected for its rapidity and reproducibility, which were essential to this study. The FRAP reagent was prepared by mixing acetate buffer, 2,4,6-Tri(2-pyridyl)-s-triazine (TPTZ) solution, and ferric chloride solutions in a 10:1:1 ratio. Extracts and standards (10 μL) were measured in microcentrifuge tubes. FRAP reagent (300 μL) was added to the content of the microcentrifuge tubes and vortexed. The content of each tube (100 μL) was transferred to a Nunc 96-well plate. Absorbance was measured immediately (3–5 min from plating) in a GENios TECAN plate-reader at 595 nm. Ascorbic acid (Sigma-Aldrich, Darmstadt, Germany) standards with concentrations ranging from 10 to 1000 μmol/L were used to generate standard plots ($R^2 \geq 0.99$) and an equation was used to calculate the antioxidant capacity of extracts as compared to ascorbic acid concentrations. The results were expressed in grams of ascorbic acid equivalent per litre (g AAE/L).

2.5.2. Physical Tests

The colour was measured using a CT-1100 ColorQUEST HunterLab taking measurements for transmittance. Standard black plates were used for calibration. The L*, a*, and b* readings were obtained and used to calculate the chroma and hue angles using the following equations:

$$Hue\ angle\ (^o) = \left(\frac{b^*}{a^*}\right) \tag{6}$$

$$Chroma = \left[(a^*)^2 + (b^*)^2\right]^{\frac{1}{2}} \tag{7}$$

The hue angle and chroma were used on a CIE 1979 L* *a* *b* colorimetric system diagram to identify the colour and monitor any changes. Chroma was presented in absolute values ranging from 0 to 100, while the unit for the hue angle was degrees (ranging from 0 to 360).

The pH was measured using a pH meter (Mettier-Toledo SevenEasy) calibrated using buffer solutions 4.0 and 7.0 (Sigma-Aldrich, Darmstadt, Germany).

### 2.6. Statistics

Three batches of Roselle were prepared, and each batch was analysed in triplicate. Statistical analyses were carried out using linear mixed models for repeated measures on SPSS Statistical Software (IBM, Chicago, IL, USA, Version 24) to compare groups based on sweeteners (*n* = 2) and spice flavour (*n* = 2). In addition, ANOVA and Tukey's tests were used to compare means using SPSS. Results with $p < 0.05$ (95% confidence level) were considered significantly different. A summary of the statistical analysis between treatments is shown in Appendix A (Tables A1–A3). Three-way ANOVA (XLSTAT version 2017.1) was used to determine the effect of the day (*n* = 2), sweetener (*n* = 2), and spice type (*n* = 3) on properties of interest for the beverages (*n* = 9). Results with $p < 0.05$ (95% confidence level) were considered significantly different.

## 3. Results and Discussions

### 3.1. Anthocyanins

#### 3.1.1. Kinetics of Anthocyanins Degradation

Storage of the Roselle beverages at room temperature (21 °C) for 90 days resulted in anthocyanin degradation, which followed a first-order rate of reaction (Figure 1). Scatter plots of the anthocyanin degradation over the 90-day storage period are shown in Figure 1, while the kinetic data for the Roselle anthocyanin degradation are shown in Table 1.

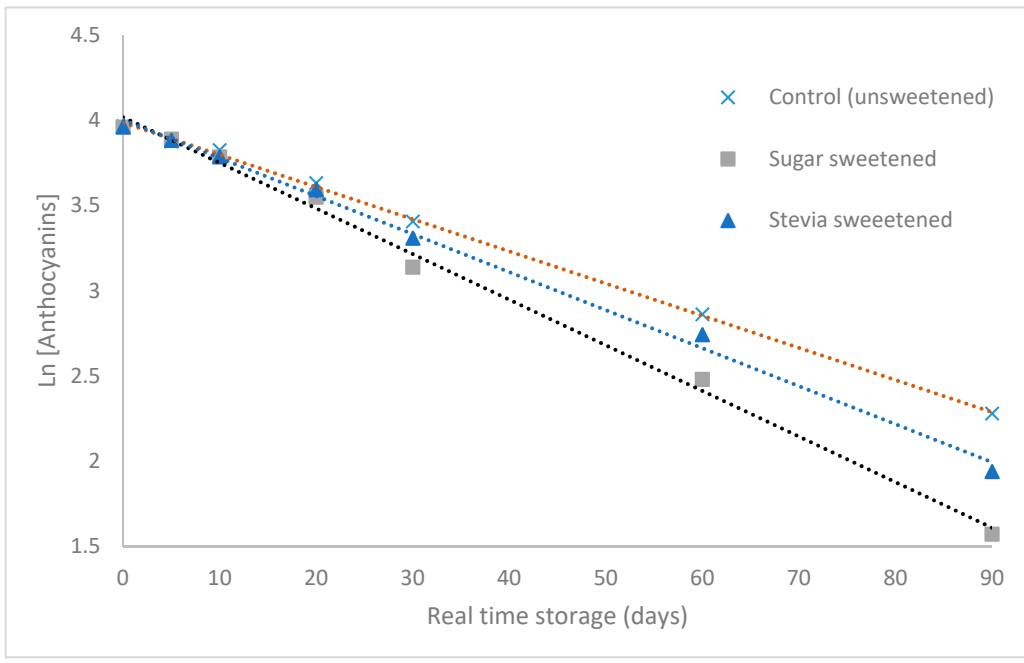

**Figure 1.** Anthocyanin degradation in Roselle beverages over 90 days of storage at 21 °C.

**Table 1.** Summary of kinetic data on anthocyanins from a 90-day real-time study with unsweetened and sweetened Roselle beverages.

| Beverage | Rate Constant (Day$^{-1}$) $\pm$ SD | Coefficient of Determination ($R^2$) $\pm$ SD | Half-Life $t_{1/2}$ (Days) $\pm$ SD |
|---|---|---|---|
| Control (unsweetened) | 0.0181 $\pm$ 0.0039 [a] | 0.9774 $\pm$ 0.0325 | 39 $\pm$ 7 [a] |
| Sugar-sweetened | 0.0240 $\pm$ 0.0048 [a] | 0.9830 $\pm$ 0.0226 | 30 $\pm$ 7 [a] |
| Stevia sweetened | 0.0208 $\pm$ 0.0025 [a] | 0.9908 $\pm$ 0.0043 | 34 $\pm$ 4 [a] |

Different superscript letters are used to designate values that are statistically significant ($p < 0.05$).

The rate constant of the anthocyanins in the unsweetened beverage (control) was lower than in the beverages containing sugar and Stevia (Table 1). However, the apparent

differences in the stability of the Roselle anthocyanins using both sweeteners are statistically insignificant ($p > 0.05$). This indicates that sweeteners (and especially promising, the Stevia option) can be used without concern for the immediate and storage quality attributes (especially anthocyanins) of Roselle beverages.

3.1.2. Accelerated Study on Anthocyanins

Using the data in Tables 1 and 2 and applying Equation (4), it was found that one day under accelerated conditions (at 40 °C) was equivalent to approximately six days under real-time conditions (room temperature, 21 °C) for the control beverage and 5 days for the sweetened beverages without spices.

**Table 2.** Total anthocyanin kinetics data for unsweetened (control), sweetened, and spice-flavoured Roselle beverages ($n = 9$) stored under accelerated conditions (40 °C) for 30 days.

| Sample | Initial Total Anthocyanin * (mg C3GE/L) | Rate Constant (Day$^{-1}$) | Coefficient of Linear Correlation ($R_2$) | Half-Life $t_{1/2}$ (Days) |
|---|---|---|---|---|
| Control | 34 ± 4 [a] | 0.1022 | 0.9811 | 7 [b] |
| Unswt_cinnamon | 31 ± 4 [a] | 0.0532 | 0.9652 | 13 [a] |
| Unswt_ginger | 32 ± 4 [a] | 0.1095 | 0.9622 | 6 [b] |
| Sugar_no spice | 34 ± 8 [a] | 0.1295 | 0.9775 | 5 [b] |
| Sugar_cinnamon | 35 ± 9 [a] | 0.0334 | 0.8132 | 21 [a] |
| Sugar_ginger | 34 ± 8 [a] | 0.1479 | 0.9303 | 5 [b] |
| Stevia_no spice | 35 ± 9 [a] | 0.106 | 0.9773 | 7 [b] |
| Stevia_cinnamon | 33 ± 9 [a] | 0.0575 | 0.9153 | 12 [a] |
| Stevia_ginger | 33 ± 10 [a] | 0.1073 | 0.9373 | 6 [b] |

* Total anthocyanin measured in cyanidin 3-glucoside equivalent. Standard deviation follows the ± sign. Different superscript letters are used to designate values that are statistically significant ($p < 0.05$).

As shown in Table 2, the initial anthocyanin content was not significantly altered by the inclusion of either sweetener or spice ($p > 0.05$).

Over the period of accelerated storage, the sweeteners did not impact the stability of the anthocyanins. Consequently, there were no significant differences in the reaction rates or half-lives of the spice-free beverages (Table 2). This validates the results from the real-time storage study. However, it varies from the results obtained by Tsai et al. [5], where using 20–60% sugar produced a decrease in the degradation of anthocyanins with increasing sugar content and, correspondingly, increased the half-lives of the anthocyanins. However, it must be noted that the sugar concentrations were much higher in that study (i.e., 20% versus the 8% used in this study); thus, the observed changes are potentially due to the dilution or reduction in water activity rather than any reactions with the anthocyanins.

The inclusion of cinnamon in the beverages displayed a stabilising effect on the anthocyanins over the storage period. This effect was significant ($p < 0.001$), even in the absence of sweeteners ($t_{1/2}$ increased from 7 to 13 days). However, in combination with sugar, the effect was quite pronounced. Moreover, the rate of anthocyanin degradation was reduced (Table 1) and the half-life significantly increased up to 21 days in the cinnamon–sugar combination.

Several compounds contained in the cinnamon extract may be responsible for this beneficial effect on the anthocyanins. They include catechin, protocatechuic acid, proanthocyanidins, and cinnamaldehyde. Several studies have reported the ability of these compounds and their reaction products to stabilise anthocyanins through co-pigmentation reactions [8]. Cinnamaldehyde is slowly oxidised to form cinnamic acid, while studies have shown that anthocyanins are more stable when acylated with cinnamic acid [22]. It is not clear what the mechanism for the stability of the total anthocyanins is and particularly the

role of the sugar, which seems to complement stability. However, considering the increase in total phenolic content in sugar-sweetened beverages, phenolics contained in cinnamon are potentially reacting with the reducing sugars. Several studies revealed better stabilising effects of glucose on anthocyanin compared to sucrose or fructose and demonstrated a co-pigmentation reaction between glucose and chlorogenic acid [10,23]. Perhaps in this current study, the reaction is between glucose and any of the above-listed phenolic compounds introduced with the inclusion of cinnamon.

Conversely, ginger did not positively impact the anthocyanin stability of the beverages, which is particularly highlighted in Table 2, where the half-lives of all treatments containing ginger were lower than the control.

### 3.2. Total Phenolic Content

The addition of sweeteners resulted in no significant changes ($p = 0.06$) to the initial total phenolic content of the Roselle beverages (Figure 2). Furthermore, the addition of sweeteners did not significantly affect the degradation of the phenolic compounds during their storage. Alternatively, the addition of spices to the sweetened beverages contributed to an initial rise in the total phenolic content of the beverages, although it was not statistically significant.

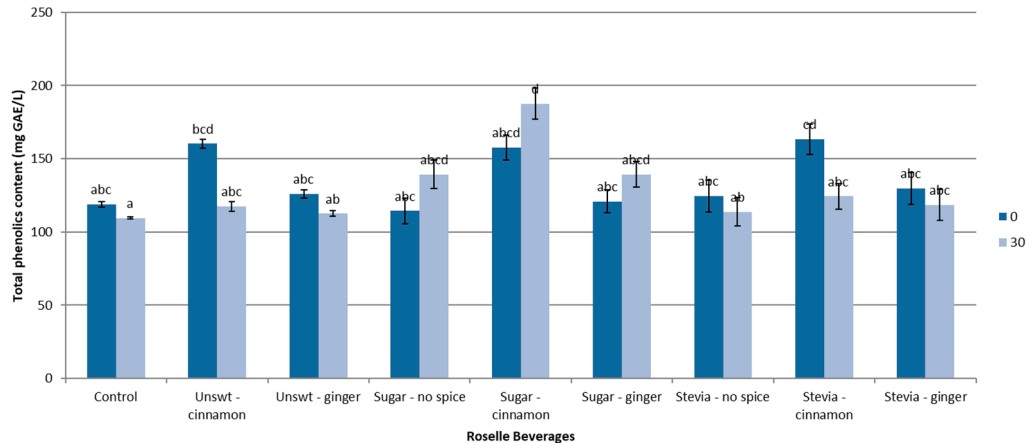

**Figure 2.** Total phenolic content (mg GAE/L) profile of the unsweetened or sweetened Roselle beverages (with sugar or Stevia Reb A) and with or without spice (cinnamon or ginger) on days 0 and 30 of the accelerated storage at 40 °C. Each bar/point represents mean values ± SD. Symbols a–d denote the level of differences between treatments.

With the initial additions of only sugar and Stevia (on Day 0), no significant increase in total phenolic content was observed. However, following the 30 days of accelerated storage at 40 °C, the total phenolic content of the sugar-sweetened Roselle beverages significantly increased. However, this result was most likely due to sucrose hydrolysing in the presence of acids to produce reducing sugars, which are known to interfere with the Folin–Ciocalteu reagent. Thus, after correcting for this error, there were no significant effects from the spices on the stability of the total phenolic content in the sugar-sweetened beverages. Similarly, Stevia did not alter the stability of the phenolic content during the storage period. This finding is in contrast to conclusions presented by Perez-Ramirez, Castano-Tostado, Leon, Rocha-Guzman, and Reynoso-Camacho [24], who observed improved stability on specific phenolic compounds, including gallic acid, quercetin, and rosmarinic acid after the addition of Stevia (97% purity; 14–15 g/L), although the Stevia quantities used in that study were much higher than this present study (only 0.32 g/L of Stevia Reb A was used).

With the addition of cinnamon to the control and sweetened bases, the corresponding observed increase in total phenolic content was because of the additive effect of the phenolic compounds in the spices. Cinnamon increased the initial total phenolic content more than ginger because it contains over 10 times more phenolics than ginger [25].

### 3.3. FRAP Activity

The initial FRAP activity did not significantly change ($p = 0.944$) with the addition of either the sweeteners or spices (Figure 3). However, there was a significant increase ($p < 0.039$) in initial antioxidant activity when cinnamon was added to the unsweetened or sweetened Roselle beverages. Over the storage period, the antioxidant activity decreased for all treatments (with or without spices). Across the sweetened and unsweetened Roselle beverages, cinnamon reduced the degradation of antioxidant activity, which corresponded to the stabilisation effect on the anthocyanins (Table 2). Moreover, this effect was significant in the sugar-sweetened beverages (Figure 3), where adding cinnamon led to the least antioxidant activity loss. Alternatively, the addition of ginger to the sweetened or unsweetened Roselle beverages had no obvious effect on the antioxidant activity. The findings in the Stevia-sweetened beverages are similar to previous findings by Korir, Wachira, Wanyoko, Ngure, and Khalid [26], where adding 3 g/L of Stevia did not improve the antioxidant capacity of black tea. However, the results in the literature are contradictory for the addition of sugar. Indeed, when 30 g/L of sugar was added to the black tea the antioxidant activity was reduced [26]. Conversely, the FRAP antioxidant activity of mulberry extract increased with the addition of sugar (20–60%), although this was alongside the presence of heating according to Tsai et al. [5]. In the study by Korir et al. [26] the mechanism of the reduction in antioxidant capacity was postulated to be because of glucose–gallic complexes. In the study by Tsai et al. [5], the increase in FRAP antioxidant activity was attributed to products of the Maillard reaction, which have been verified in other studies to be effective antioxidants [27].

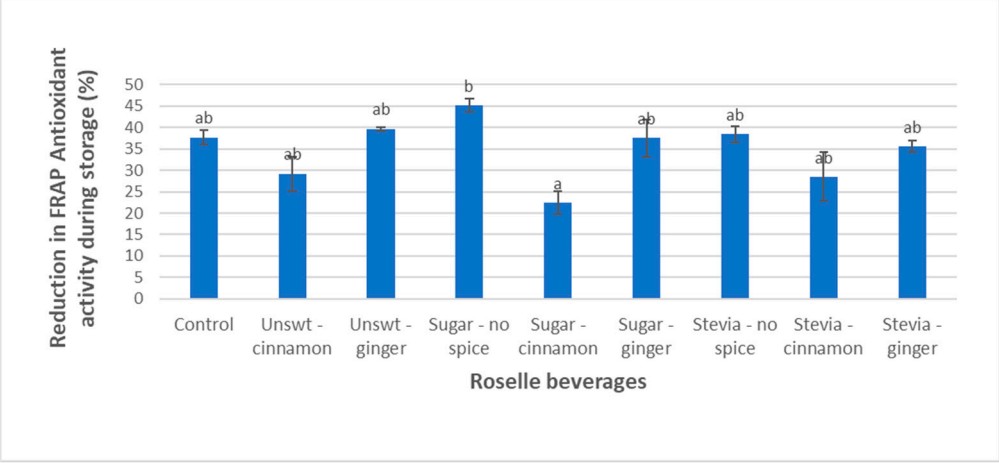

**Figure 3.** Percentage loss in FRAP antioxidant activity in the Roselle beverages between days 0 and 30 of the accelerated storage at 40 °C. Each bar/point represents mean values + SD. Symbols a and b denote the level of differences between treatments.

Furthermore, the sugar–cinnamon combination, which induced the highest protective effect on FRAP activity, corresponds to the highest protective effect on anthocyanins with the same combination (Table 2). This finding agrees with those by Tsai et al. [28] on model Roselle beverages, in which the monomeric anthocyanins positively correlate with the FRAP activity (coefficient 0.97).

### 3.4. Colour

The chroma of the Roselle beverages decreased with an increase in storage time (Figure 4a). The sweetener type did not affect the initial colour of the Roselle beverages and there was no subsequent effect on the stability of the colour during storage. Although there was no initial effect of the spices on the chroma of the beverage, the inclusion of cinnamon in the unsweetened and sweetened beverages led to a decrease in the chroma over the storage period, particularly within 15 days. Eventually, the chroma in all the Roselle

beverages on day 30 of the accelerated study was the same, except in those where sugar was used in combination with the cinnamon. For the latter, there was a significant effect of colour chroma (Figure 4a), which is likely linked to the improved stability of the anthocyanins in the beverages containing the sugar–cinnamon combination. For all beverage iterations, the correlation between the total anthocyanin content and chroma was high ($R^2$ value = 0.97 $\pm$ 0.02; R = 0.99).

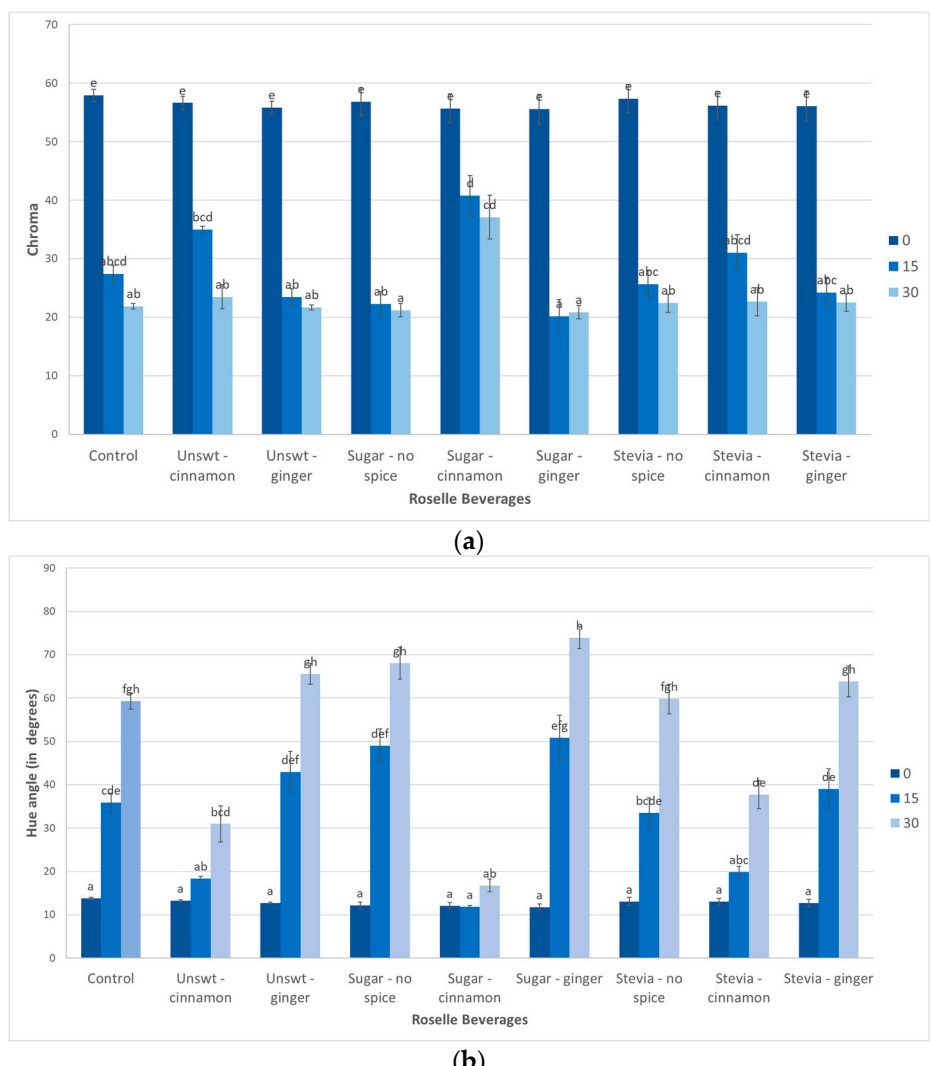

**Figure 4.** (**a**) Chroma profile of the unsweetened or sweetened Roselle beverages (with sugar or Stevia Reb A) and with or without spice (cinnamon or ginger) on days 0, 15, and 30 of the accelerated storage at 40 °C. (**b**) Hue angle profile of the unsweetened or sweetened Roselle beverages (with sugar or Stevia Reb A) and with or without spice (cinnamon or ginger) on days 0, 15, and 30 of the accelerated storage at 40 °C. Each bar/point represents mean values + SD. Symbols a–h denote the level of differences between treatments.

The colour hue angle (Figure 4b) was less correlated (negatively) to the total anthocyanins for all the beverage iterations ($R^2$ value = 0.67 $\pm$ 0.05; R = −0.82). In addition, there was no effect of the sweetener on the hue angle, although the type of spice added did have an effect. All beverages containing cinnamon had a lower hue angle (redder) throughout storage. The sugar–cinnamon beverage with the lowest hue angle at day 30 in the accelerated storage group maintained the reddest colour, thus, linking well with the anthocyanin data. Whereas, the hue angle of the other beverages increased, tending towards a more yellow colour.

Overall, the sugar–cinnamon correlation resulted in the highest stability of colour (chroma), which agrees with the highest anthocyanin protective effect found for this beverage combination. This agrees with the findings by Tsai et al. [28], who found that monomeric anthocyanins in a model Roselle beverage were mainly responsible for the red colour and were positively correlated with the colour density (R 0.872).

### 3.5. pH

For most of the Roselle beverages, the pH values did not change significantly throughout the storage period, as shown in Figure 5. The addition of sweeteners or spices did not significantly alter the initial pH values. For all beverages, the pH was reduced by day 15, yet was increased by day 30. This suggests changes occurring in the constituents of the phenolic compounds during storage. The pH did not correlate with any of the other measured parameters.

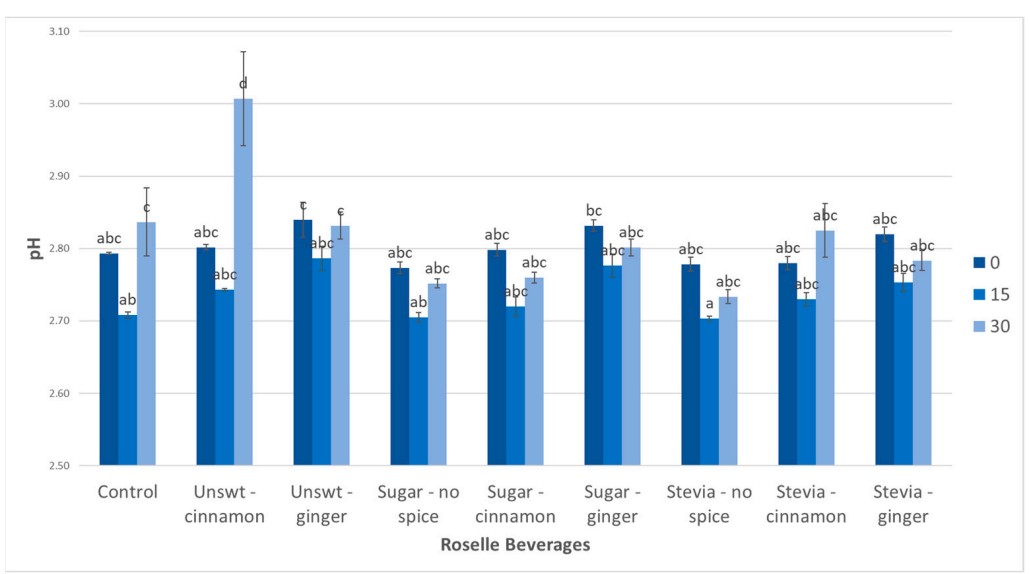

**Figure 5.** The pH profile of the unsweetened or sweetened Roselle beverages (with sugar or Stevia Reb A) and unflavoured or flavoured (with cinnamon or ginger) on days 0, 15, and 30 of the accelerated storage at 40 °C. Each bar/point represents mean values + SD. Symbols a–d denote the level of differences between treatments.

### 4. Conclusions

As little as 1 g/L cinnamon added to a Roselle beverage (especially one that had been sugar-sweetened) showed some promising results as a stabilising ingredient for anthocyanins, antioxidant activity, and for the Roselle beverage colour. Although replacing sugar (80 g/L) with Stevia Reb A (Stevia, 0.32 g/L) did not have similar stabilising effects upon reaction with cinnamon, the Stevia–cinnamon combination still showed promising results for anthocyanin and colour stability. It may be helpful to investigate a dose-response effect upon combining cinnamon and different types, or higher quantities of Stevia, to maximize this potential. Although ginger (1 g/L) did not perform as well as cinnamon in the Roselle beverages, it may be helpful to investigate higher concentrations of this spice to eliminate the possibility that low concentrations contributed to the lack of results.

The mechanisms of interactions between these spices, sweeteners, and anthocyanins also need further exploration. Other congruent spices (such as basil, cloves, etc.), which are typically (or can be) used in Roselle beverages should be investigated, particularly for their anthocyanin stability potential. The effect of cinnamon on other anthocyanin beverage matrices should be verified in the interest of clean-label beverage quality stabilities.

**Author Contributions:** Conceptualization, E.D.O., P.J. and N.H.; methodology, E.D.O., P.J. and N.H.; software, E.D.O.; validation, P.J. and N.H.; formal analysis, E.D.O., P.J. and N.H.; investigation, E.D.O.; resources, E.D.O., P.J. and N.H.; data curation, E.D.O., P.J. and N.H.; writing—E.D.O.; writing—review and editing, P.J. and N.H; visualization, E.D.O.; supervision, P.J. and N.H.; project administration, P.J.; funding acquisition, E.D.O. All authors have read and agreed to the published version of the manuscript.

**Funding:** This research was funded by a combination of funds from Niger Delta Development Commission: NNDC/DEHSS/2013PGFS/EDO/2; Schlumberger Foundation: Faculty for the Future program and The Gen Foundation (Registered UK Charity No. 1071026).

**Institutional Review Board Statement:** Not applicable.

**Informed Consent Statement:** Not applicable.

**Data Availability Statement:** Not applicable.

**Conflicts of Interest:** The authors declare no conflict of interest.

## Appendix A

**Table A1.** Type III Tests of Fixed Effects for Total Phenolics Content (from SPSS).

| Source | Numerator df | Denominator df | F | Sig. |
|---|---|---|---|---|
| Intercept | 1 | 278.822 | 25,178.241 | 0.000 |
| sweetener | 2 | 278.822 | 39.277 | 0.000 |
| spice | 2 | 278.822 | 164.626 | 0.000 |
| sweetener * spice | 4 | 277.706 | 5.134 | 0.001 |
| Day | 12 | 39.903 | 2.296 | 0.025 |
| sweetener * Day | 24 | 39.903 | 1.739 | 0.060 |
| Day * spice | 24 | 39.903 | 0.529 | 0.950 |

Dependent Variable: TP. "*" to indicate a combination of treatments.

**Table A2.** Type III Tests of Fixed Effects for Total Anthocyanins (from SPSS).

| Source | Numerator df | Denominator df | F | Sig. |
|---|---|---|---|---|
| Intercept | 1 | 173.482 | 4535.259 | 0.000 |
| sweetener | 2 | 173.482 | 1.048 | 0.353 |
| spice | 2 | 173.482 | 37.617 | 0.000 |
| sweetener * spice | 4 | 166.017 | 13.917 | 0.000 |
| Day | 12 | 55.728 | 149.595 | 0.000 |
| sweetener * Day | 24 | 55.728 | 0.765 | 0.761 |
| Day * spice | 24 | 55.728 | 2.316 | 0.005 |

Dependent Variable: AnCy.

**Table A3.** Type III Tests of Fixed Effects for Antioxidant Capacity (FRAP, from SPSS).

| Source | Numerator df | Denominator df | F | Sig. |
|---|---|---|---|---|
| Intercept | 1 | 226.260 | 10,547.282 | 0.000 |
| sweetener | 2 | 226.260 | 15.606 | 0.000 |
| spice | 2 | 226.260 | 61.105 | 0.000 |

**Table A3.** *Cont.*

| Source | Numerator df | Denominator df | F | Sig. |
|---|---|---|---|---|
| sweetener * spice | 4 | 235.015 | 2.564 | 0.039 |
| Day | 12 | 39.798 | 13.519 | 0.000 |
| sweetener * Day | 24 | 39.798 | 0.374 | 0.994 |
| Day * spice | 24 | 39.798 | 0.540 | 0.944 |

Dependent Variable: AnOX.

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
