# Peer review of "The Effect of Cinnamon and Ginger Spices on Anthocyanins in Sweetened Roselle Beverages"

_beverages, doi:10.3390/beverages9010024_

Round 1
Reviewer 1 Report
The topic of the paper is very interesting since currently, people look for options of beverages with high nutritional value mainly related to bioactive compounds such as anthocyanins due to health benefits. However, the paper should be improved in the analysis of the results obtained in each parameter measured. As the information presented is part of a doctoral thesis of 2017, I consider it necessary to update the paper references since there is new information associated mainly with the subject of acylation of anthocyanins, and this will allow improving the quality of the paper.
Here are some general recommendations:
*Line 85, it is necessary to include the word "and" before the number 2
*Line 100. It is necessary to clearly indicate whether a follow-up of the behavior of the drink was carried out during the 30 days or if it is only a punctual evaluation (days 0 and 30) since results are shown in color at 0, 15, and 30 days. And also how the values of K for anthocyanins were calculated if it was a punctual evaluation.
*Line 104 to 120, Rewrite to give greater clarity since not all the data of the thesis from which this information was extracted is known. So you have to be more specific.
*Line 136. check that the units of 1000 are correctly written
*Line 137. Indicate which was the factor conversion of cyanidin-3-glucoside to delphinidin 3-sambubioside (D3S). It is not clear
*Line 155. The technique of FRAP required time of reaction.
*Line 196 - Chemical kinetics. I consider that the information about real-time shelf life should be shown and it could be in the supplementary information. This is important for understanding calculus.
*Line 208 to 211. It indicates that the rate of reactions was for the parameters of total phenolic content, anthocyanins, and FRAP antioxidant, and in the results only show data for anthocyanins. Check
*Line 227. As a suggestion, show the results of the analysis in the same order as indicated in the methodology
*In figure 1, change sugar - no cinnamon to Sugar without spice, and stevia no cinnamon to Stevia without spice
*Line 246 - At compared to the samples control and sugar no cinnamon did not show an increase. Check
*Line 246 to 255. The analysis of the results and the effect of sugar and spices are not clear.
*Line 263, it is indicated that no statistical difference but when checking the data if there is a difference (34, 31 y 32).
*Line 263. The sweeteners do not change the value of the anthocyanins but the species reduce the value.
*Table 1 - It is necessary to include statistical analysis
*Line 274. Which samples correspond to the unflavoured
*Line 282. And which is the effect of ginger?
*Line 302. It did not mention the effect of time on the reduction of antioxidant activity.
*Line 327. The information indicated in this sentence must be after the analysis of the color behavior. Additionally, this correlation of 0.99 corresponds to the data for time 0, 15, and 30 days.
*Line 328. it is important to indicate that the color is expressed as chroma and Hue
*Line 335. This change significantly in chroma value in all samples (60 to 20) could be perceived by the eye human. Since it is important to have beverages have a balance between anthocyanins content, AA, and sensory quality (in this case color). And this 50% decrease is not analyzed
*Check the name of each treatment in all figures and tables since different nomenclature is used.
*It could include these data in the table 1
*Line 354. Rewrite the conclusions because the text is more associated with the analysis of the results than with the conclusions of the research and it is important to strengthen recommendations for future research on the topic.
Reviewer 2 Report
I would like to thank the authors for their interesting work entitled “Cinnamon and Ginger spices effect on the anthocyanins in sweetened Roselle beverage” that is quite a contribution in the field. Their work which investigated the effect of cinnamon and ginger on anthocyanin and related properties is methodologically sound. However, there are few issues that should be considered in their revised version
1. Results L239: Under the sub section total phenolic content you have stated addition of sweeteners have no significant effect at all the stages. However, the effect of adding spices have resulted in rise of total phenolic content particularly the rise is statistically significant on addition of cinnamon. Can you please include the appropriate p-value instead of writing (p<0.05)? at least where the result is significant. This will make your work statistically reliable and reproducible.
2. Results L264: The sentence “……spices are devoid of anthocyanins….” is misleading since some studies claim to have identified anthocyanins in both ginger and cinnamon despite their quantity. Please reconsider using this claim to justify your results. I believe the second reason will suffice.
3. Table 1: The rate constant for SRA_no spice is the least but the half-life is 7 days. Please check it again.

Round 2
Reviewer 1 Report
the new version of the manuscript was significantly improved and this version included several comments indicated in the evaluation, so I considered that it could be accepted for publication.
Author Response
Thank you very much indeed for the comments and feedback. Some further improvements were made line with the academic editors comments, including:
- Included statistical superscript letters to Figures 4 and 5.
- Replaced Figure 3 (and included statistical superscript letters) due to difficulties with the describing the results of the Antioxidant activity of the beverages in line with the previous figure and statistical data (as explained in our previous letter).
- Revised related text to ensure that discussions are congruent.
We hope these changes further assist with presenting and discussing the data more clearly.